# In Situ H-Radical Surface Treatment on Aluminum Gallium Nitride for High-Performance Aluminum Gallium Nitride/Gallium Nitride MIS-HEMTs Fabrication

**DOI:** 10.3390/mi14071278

**Published:** 2023-06-21

**Authors:** Yannan Yang, Rong Fan, Penghao Zhang, Luyu Wang, Maolin Pan, Qiang Wang, Xinling Xie, Saisheng Xu, Chen Wang, Chunlei Wu, Min Xu, Jian Jin, David Wei Zhang

**Affiliations:** State Key Laboratory of ASIC and System, School of Microelectronics, Fudan University, Shanghai 200433, China; 20112020039@fudan.edu.cn (Y.Y.); 20212020129@fudan.edu.cn (R.F.); phzhang19@fudan.edu.cn (P.Z.); wangly20@fudan.edu.cn (L.W.); 21112020100@m.fudan.edu.cn (M.P.); 21112020111@m.fudan.edu.cn (Q.W.); 21212020013@m.fudan.edu.cn (X.X.); ssxu@fudan.edu.cn (S.X.); wuchunlei@fudan.edu.cn (C.W.); 22112020096@m.fudan.edu.cn (J.J.); dwzhang@fudan.edu.cn (D.W.Z.)

**Keywords:** AlGaN/GaN, in situ H-radical treatment, C–V frequency dispersion, pulse-mode stress, interface traps, MIS-HEMTs, current collapse, dynamic on-resistance

## Abstract

In this work, we demonstrated a low current collapse normally on Al_2_O_3_/AlGaN/GaN MIS-HEMT with in situ H-radical surface treatment on AlGaN. The in situ atomic pretreatment was performed in a specially designed chamber prior to the thermal ALD-Al_2_O_3_ deposition, which improved the Al_2_O_3_/AlGaN interface with $D_{it}$ of ~2 × 10^12^ cm^−2^ eV^−1^, and thus effectively reduced the current collapse and the dynamic $R_{on}$ degradation. The devices showed good electrical performance with low $V_{th}$ hysteresis and peak trans-conductance of 107 mS/mm. Additionally, when the devices operated under 25 °C pulse-mode stress measurement with $V_{DS,Q}$ = 40 V (period of 1 ms, pulse width of 1 μs), the dynamic $R_{on}$ increase of ~14.1% was achieved.

## 1. Introduction

The third-generation semiconductor gallium nitride (GaN) has a large band gap and high electron saturation rate, making it particularly suitable for the preparation of high switching speed, portable and miniaturized power devices [1,2]. In recent years, GaN power devices have been highly demanded in the consumer electronics field, such as fast chargers and adapters, and are making inroads into data centers, LiDAR, wireless charging and photovoltaics [3,4]. However, GaN devices face a number of scientific and technical challenges that seriously hinder their industrial application and the full realization of their market value [5].

One of the pressing problems is the current collapse caused by charge trapping, known as increasing dynamic $R_{on}$ in the case of power switch devices [6,7]. When the device is in the off-state condition, electrons are captured by defects to form negative charge centers, and the capture position is usually on the drift region from the side of the gate close to the drain [8,9]. This degradation widely occurred in both E-mode and D-mode GaN HEMT devices [10,11]. When the device is switched from off-state to on-state conditions, electrons are emitted by defects and participate in the process of conduction again. However, this will lead to the increase in conduction loss, which will reduce the efficiency of the system [12]. Currently, several trapping mechanisms responsible for current collapse and increasing dynamic $R_{on}$ have been investigated, and the discussion for the sources of defects is mainly focused on two directions [13]. One is the surface traps of AlGaN [14], and the other is the traps introduced by carbon doped in the epitaxial growth of GaN substrates [15]. In fact, it has been reported that the high density of border traps at the surface of GaN correlated to the content of oxygen may even reach $10^{21}\;{cm}^{-3}\;{eV}^{-1}$ [16]. Many studies have confirmed that exposure of AlGaN surface to air produces a layer of structurally unstable oxides, resulting unsaturated valence bonds as the main source of AlGaN surface states [17,18,19]. Additionally, it has been reported that the removal or transformation of Ga-O can greatly reduce C–V frequency dispersion and improve the device performance [20,21]. Although many wet processes were developed to sufficiently remove the Ga-O layer [22,23], the vacuum break between wet clean and surface passivation inevitably causes re-oxidation of the AlGaN surface which was not controllable in practice. Additionally, the dry processes in plasma cause surface damage more or less [24]. So far, there is no such process that can effectively reduce the surface traps of AlGaN without causing obvious damage and additional contamination.

H-radical treatment using electrically neutral hydrogen radicals, also known as atomic hydrogen or simple H*, is a procedure that removes contaminant layers or oxides by chemically reacting with the highly reactive valence electrons of the hydrogen radicals [25]. It is well known that H radicals are highly reactive reducing agents that react rapidly with the surface molecules to effectively remove the oxide layer from the material surface and activate the surface properties [26]. In this work, in situ H-radical surface treatment was developed to eliminate and inhibit the formation of Ga-O bonds on AlGaN surface in an in-house designed chamber. The RF power used for hydrogen ionized and glow was 800 W, and the RF power of the bias electrode was 0. The plasma was generated and passed through a specially designed filter which only allowed electrically neutral hydrogen radicals to enter the reaction chamber and achieved a non-plasma but radical surface treatment. The subsequent ALD dielectric deposition was integrated into a cluster so that there was no vacuum break between H-radical treatment and passivation.

We fabricated Al_2_O_3_/AlGaN/GaN MIS-capacitors and MIS-HEMTs to evaluate the developed H-radical surface treatment process, by analyzing the CV curve dispersion of MIS-capacitors and dynamic performance degradation of MIS HEMTs.

## 2. Device Characterization

The devices were fabricated on AlGaN/GaN epi-structure composed of the following layers from top to down: 20 nm Al_0.25_Ga_0.75_N barrier layer, 1 nm AlN spacer layer, 400 nm i-GaN layer, 5 μm GaN buffer layer and AlGaN nucleation layer, Si substrate. Hall measurements yielded a sheet carrier density of $1.15\times 10^{13}\;{cm}^{-2}$ and an electron mobility of $1510\;{cm}^{2}/V\cdot s$, and a sheet resistance of 360 Ω/sq. The Al_2_O_3_/AlGaN/GaN MIS-capacitors and HEMTs were fabricated, with structures illustrated in Figure 1. The process flow started from wet cleaning by acetone and isopropyl alcohol, three samples with different surface treatments were prepared: (1) S1: as-grown; (2) S2: 1 min wet etching in 1:1 hydrochloric acid; (3) S3: 1 min wet etching in 1:1 hydrochloric acid followed by 30 s H-radical treatment at 300 °C. In addition, the samples were exposed to several cycles of trimethyl aluminum (TMA) pulses before the formal ALD growth. Then, Al_2_O_3_ films with nominal thicknesses of 25 nm were deposited by thermal ALD at 300 °C with TMA as the precursor and H_2_O as the oxidant source. It is worth mentioning that the H-radical treatment and ALD deposition were performed on a cluster platform with no vacuum break. Post-dielectric annealing was carried out in an N_2_ ambient at 550 °C for 40 s [27,28]. Then, mesa-isolation was performed by Inductive Coupled Plasma (ICP) etching. In order to form good ohmic contact, the barrier layer in S/D region were recessed by Atomic Layer Etching (ALE) process with BCl_3_/Ar chemistry. After an approximate 40 nm recess etching, the samples were dipped in diluted-HCl solution, and metal stacks of Ti(10 nm)/Al(100 nm)/Ti(30 nm)/Au(40 nm) were immediately deposited by electron beam evaporation and lifted-off by ultrasonic method. Considering the possible Al_2_O_3_ crystallization [29], a low-temperature ohmic-contact process was developed at 550 °C for 60 s in N_2_ ambient. The ohmic contact resistance of ~1.1 Ω·mm was derived using the transmission line method. Finally, the devices were completed by Ni/Au (20/60 nm) gate formation.

## 3. Results and Analysis

### 3.1. Capacitance–Voltage Characteristics

To understand the Al_2_O_3_/AlGaN interface, capacitance–voltage (C–V) measurements were performed on MIS-capacitors at varied frequencies (50 KHz–2 MHz) and temperatures (−50 °C–75 °C) for Samples of S1, S2, and S3. The voltage $V_{G}$ scanned from negative to positive and all C–V curves showed two slopes, in which the first slope corresponded to the accumulation of 2DEG in the channel, and the second slope corresponded to the spillover region [30]. Additionally, we defined $V_{th}$ as the formation of 2DEG in the channel, and $V_{on}$ as the gate voltage corresponding to the start of spilling [31]. The conduction band diagrams are plotted in Figure 2, when $V_{G}$ is lower than $V_{on}$, interface traps at the Fermi level are too deep, and they cannot respond to the ac signal ($V_{ac}$). Additionally, after $V_{G}$ reaches $V_{on}$, interface traps near the Fermi level are shallow enough to respond to $V_{ac}$ by capturing or emitting electrons.

The time constants for electron capture ($\tau_{c}$) and emission ($\tau_{e}$) of interface traps at the trap energy level ($E_{T}$) and the characteristic interface trap frequency ($f_{it}$) are given by [31]:(1)$$\tau_{c}=\frac{1}{v_{sat}\sigma_{n}N_{S}}$$
(2)$$\tau_{e}=\frac{1}{v_{sat}\sigma_{n}N_{C}}exp\left(\frac{E_{C}-E_{T}}{kT}\right)$$
(3)$$f_{it}=\frac{1}{2\pi \tau_{e}}$$
where $E_{C}$ is the bottom of conduction band, k is the Boltzmann constant, $v_{sat}$ is the electron saturation drift velocity, $\sigma_{n}$ is the electron capture cross-section, $N_{c}$ is the effective density of states of GaN in the conduction band, $N_{S}$ is the electron concentration at the interface.

We plotted the measured C–V characteristics of S1, S2, S3 in Figure 3. Dispersion of the second slope was a strong indicator for Al_2_O_3_/AlGaN interface quality [31,32]. At the same measurement temperature ($T_{m}$), only shallower interface states whose frequency ($f_{it}$) is higher than $f_{m}$ can respond to $V_{ac}$ and therefore require a higher $V_{G}$ voltage to raise the capacitance. In other words, a higher $f_{m}$ leads to a larger $V_{on}$, which is consistent with the measurements. In addition, from Equations (2) and (3), it can be seen that by changing $T_{m}$, traps with a wider energy band range can be detected. In Figure 3, we extracted $V_{on}$ when the capacitance reached 110% of the first plateau capacitance $C_{1}$ [18], and we marked the shift of $V_{on}$ at 50 kHz for different temperatures to demonstrate the temperature dispersions. Obviously, the frequency dispersion and the temperature dispersion of S3 are significantly suppressed compared to that of S1 and S2. Additionally, it can be inferred that S3 has fewer interface traps than S1 and S2.

Based on Shockley–Read–Hall (SRH) recombination theory relating to frequency-related traps responding to $V_{ac}$, $∆V_{on}$ measured in the frequency range $\left(f_{1},f_{2}\right)$ reflects the interface traps existing in the energy range of $\left(∆E_{trap}\left(f_{1}\right),{∆E}_{trap}\left(f_{2}\right)\right)$. The deepest trap energy level that can be detected at the measurement frequency $f_{m}$ can be expressed as [31]:(4)$$∆E_{trap}\left(f_{m}\right)=E_{C}-E_{T}=kT_{m}ln(\frac{v_{sat}\sigma_{n}N_{c}}{2\pi f_{m}})$$

Additionally, the distribution of Al_2_O_3_/AlGaN interface states can be obtained as [31,32]:(5)$$D_{it}\left(∆E_{trap}=\frac{\left(∆E_{trap}\left(f_{1}\right)+{∆E}_{trap}\left(f_{2}\right)\right)}{2}\right)=\frac{C_{ox}\cdot ∆V_{on}}{q\left(∆E_{trap}\left(f_{1}\right)-{∆E}_{trap}\left(f_{2}\right)\right)}-\frac{C_{ox}+C_{br}}{q^{2}}$$
where $C_{ox}$ is the capacitance of the ${Al}_{2}O_{3}$ dielectric, $C_{br}$ is the capacitance of the AlGaN barrier. Additionally, they can be calculated from the two plateau capacitances of the C–V curve which showed full accumulation (the part where the voltage is applied high enough to fully accumulate the capacitance is not shown in Figure 3).

We used the C–V test data (partly not shown in Figure 3) at six sets of temperatures (−50 °C, −25 °C, 0 °C, 25 °C, 50 °C, and 75 °C) to calculate the Al_2_O_3_/AlGaN interface $D_{it}$ distribution in the energy range of 0.21~0.42 eV from AlGaN conduction band. Table 1 summarized the key parameters of the calculation. As shown in Figure 4, with H-radical treatment, Al_2_O_3_/AlGaN interface was significantly improved with $D_{it}$ of ~2 × 10^12^ cm^−2^ eV^−1^. In comparison, the $D_{it}$ of S2 was ~6 × 10^12^ cm^−2^ eV^−1^ and S1 was ~2 × 10^13^ cm^−2^ eV^−1^. This result showed that a better Al_2_O_3_/AlGaN interface was obtained through H-radical surface treatment.

### 3.2. MIS-HEMT Devices Characteristics

The MIS-HEMTs DC transfer characteristics were compared for the three different surface treatments of S1, S2, and S3. Around pinch-off voltage, because of the strong polarization field in the III-N barrier layer, the Fermi level position is deep into the bandgap at the dielectric/III-N interface [31]. When $V_{GS}$ sweeps up, traps capture electrons. When $V_{GS}$ sweeps down, some relatively shallow traps can emit electrons and become empty, while other relatively deep interface states still remain filled with electrons even if they are above the Fermi level and act like “fixed charges”. In addition, the charging and discharging process of traps can be reflected from the subthreshold slope. As shown in Figure 5, samples of S1 and S2 exhibited a threshold hysteresis (Δ$V_{th}$) of ∼0.4 V and ∼0.2 V, and a subthreshold slope (SS) of ∼85 mV dec^−1^ and ∼83 mV dec^−1^, respectively, indicating that hydrochloric acid had removed most of the natural oxide layer. In contrast, a small Δ$V_{th}$ of ∼0.01 V as well as a small SS of ∼65 mV dec^−1^ was observed for the S3. This indicates that H-radical treatment can effectively suppress interface trap related switching transients in MIS-HEMTs [18,34,35]. In addition, the threshold voltages defined at drain current of 1 μA/mm are about −14, −12 and −12 V for S1, S2, and S3, respectively, which might be related to the positive charges in the natural oxide layer or other contaminants on the surface of S1 [36,37]. Note that the variation of $I_{D}-V_{DS}$ curves for S1, S2, S3 was not found to be obvious, hence output characteristics for the samples were excluded from this paper.

In order to evaluate current collapse and the dynamic $R_{on}$ degradation of GaN HEMT in a power converter-like application scenario, pulse-mode stress measurement was adopted [38]. When the device is in the off-state with $V_{DS,Q}$, the channel below the gate is depleted, but electrons would fill the traps on the Al_2_O_3_/AlGaN interface of gate-drain drift region under the influence of a lateral electric field [15], resulting in the increase in dynamic resistance [13]; when the device is in the stress-free on-state, the captured electrons are released, and the dynamic resistance is gradually reduced [39,40,41,42]. Therefore, when the device constantly switches between the two states, some defects cannot be recovered in time, thus reaching a dynamic equilibrium process. Figure 6 showed a schematic of the waveforms of the gate and drain voltages used in the measurements, with the pulse periods of 1 ms. The off-state was @$\left(V_{GS,Q}-V_{th}\right)=-2V$ and $V_{DS,Q}=0,15,20,30,40\;V$ with a pulse width of 0.999 ms, accounting for 99.9% of the period, and the on-state with pulse width of 1 μs was @$V_{GS}$ and $V_{DS}=0\;to\;15\;V$ with a step size of 0.1 V, and the rising and falling edge of the pulses were 0.1 μs. The test platform is a Keithley 4200 A Semiconductor Characterization System with pulsed gate and drain voltage waveforms provided by two ultrafast pulsed I–V PMU modules. All experimental data were based on multiple measurements verified on different fresh GaN-on-Si MIS-HEMTs with similar characteristics.

Figure 7 shows the comparison of the output characteristics of the samples under 25 °C double-pulse tests, and the improvement of the in situ H-radical treatment on the dynamic $R_{on}$ and current collapse of the device can be clearly seen. The dynamic $R_{on}$ of S1 and S2 was increased by 40.9% and 28.1% @$V_{GS}=0\;V$ when $V_{DS,Q}$ increased from 0 V to 40 V, while the dynamic $R_{on}$ of S3 was increased by only 14.1%.

We also manufactured S4 with H-radical treated time of 60 s. The statistical data of the dynamic $R_{on}$ under the four processes are shown in Figure 8, and the H-radical treated devices demonstrated lower dynamic $R_{on}$ degradation. The performances of 60 s H-radical treated samples were comparable to those under the 30 s treatment. This indicates that the surface treatment of H-radicals has become saturated and has a process window of at least 30 s.

## 4. Discussion and Conclusions

In this study, we compared the electrical properties of MIS-capacitor and GaN MIS-HEMT with three different treatments to show the effect of H-radical surface treatment. In the capacitance–voltage test of MIS-capacitors, traps satisfying $f_{it}>f_{m}$ can make the C–V curve stretch by charged and discharged, and we can detect the trap density by the differences in the measured C–V curves at different frequencies. Through Equation (2) we know that the de-trap time can be changed by heating up or cooling down the samples, so we analyzed the C–V frequency dispersions from −50 °C~75 °C, and the energy range detected can reach $E_{C}-E_{T}=0.21\sim 0.42\;eV$. The obtained $D_{it}$ distribution showed that the trap concentration of S2 was more than half lower than S1, and the trap concentration of S3 was more than half lower than S2. Additionally, the DC transfer characteristics of the fabricated GaN MIS-HEMTs also indicate that removing most of the natural oxide layer on the AlGaN surface with hydrochloric acid is conducive to reducing the defect concentration, and the H-radical treatment can further modify the surface such as removing the residual Ga-O, halogens, carbon or other contaminants that cannot be dealt with by hydrochloric acid.

In order to further study the effect of H-radical treatment on the performance of practical devices, the dynamic $R_{on}$ and current collapse of fabricated GaN MIS-HEMTs were compared under the double pulse measurement. In each period, we turned off the device for 0.999 ms then turned on the device for 1 μs to complete the test of current, so theoretically, under these test conditions, defects with a de-trap time greater than 1 μs would contribute to the decline of the dynamic $R_{on}$ and current collapse. Additionally, according to Equation (2), this corresponds to traps with an energy level deeper than $E_{C}-E_{T}=0.3\;eV$, which is partly consistent with the range in our C–V test. Therefore, the electrical test results of MIS-capacitor and GaN MIS-HEMT can lead to the conclusion that H-radical treatment can reduce the trap concentration on or near the interface of Al_2_O_3_/AlGaN, and thus improve the interface quality and device performance.

In this paper, a novel in situ H-radical surface treatment process on AlGaN was developed to improve Al_2_O_3_/AlGaN interface with lower $D_{it}$. The corresponding Al_2_O_3_/AlGaN MIS-HEMT devices were successfully demonstrated with much lower current collapse and dynamic on-resistance degradation under double pulse measurement. Additionally, the developed H-radical surface treatment has the characteristic of saturation, leading to a wide process window. Consequently, this developed atomic level surface pretreatment process could be the critical technology for high-performance AlGaN/GaN HEMT devices.

## Figures and Tables

**Figure 1 micromachines-14-01278-f001:**
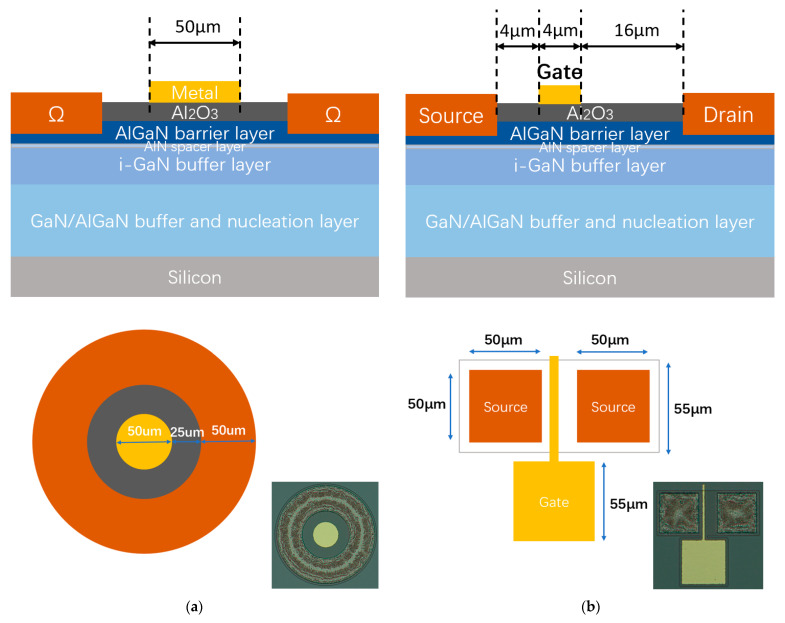
Schematic illustrations of the fabricated (**a**) Al_2_O_3_/AlGaN/GaN MIS-capacitor; (**b**) Al_2_O_3_/AlGaN/GaN MIS-HEMT.

**Figure 2 micromachines-14-01278-f002:**
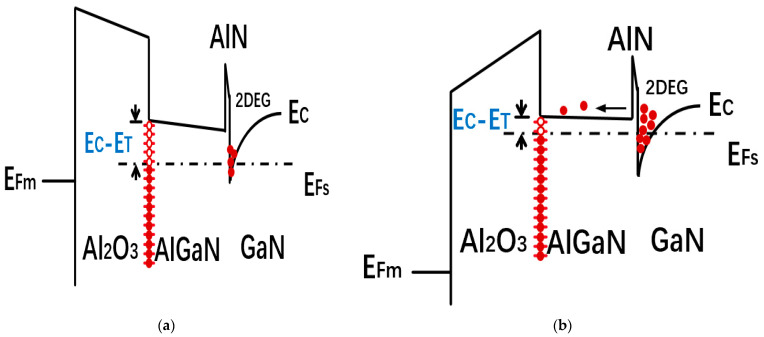
Schematic band diagrams for the Al_2_O_3_/AlGaN/GaN gate stack [31] of (**a**) accumulation region ($V_{th}<V_{G}<V_{on}$); (**b**) spillover region ($V_{G}\geq V_{on}$).

**Figure 3 micromachines-14-01278-f003:**
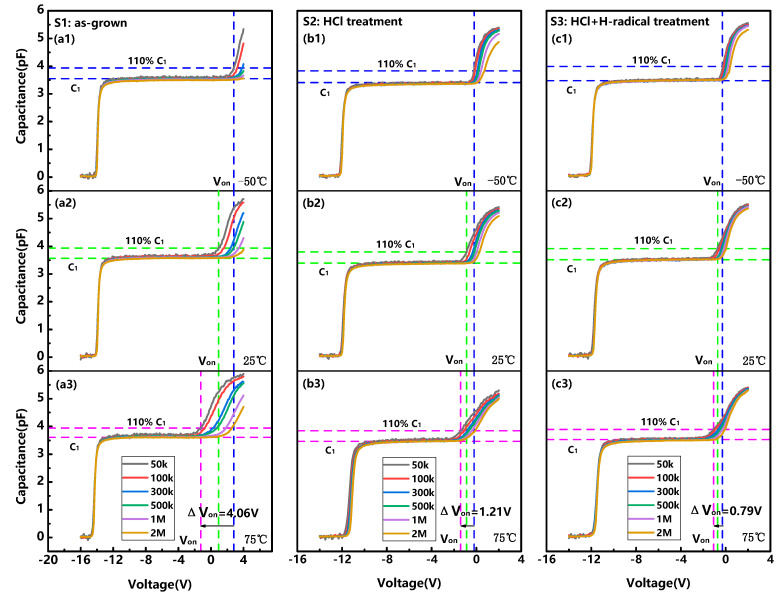
Capacitance–voltage (CV) characteristics of the MIS-capacitors (**a**) S1: as-grown; (**b**) S2: HCl treatment; (**c**) S3: HCl+ H-radical treatment at varied frequencies (50 KHz–2 MHz) and temperatures (−50 °C–75 °C).

**Figure 4 micromachines-14-01278-f004:**
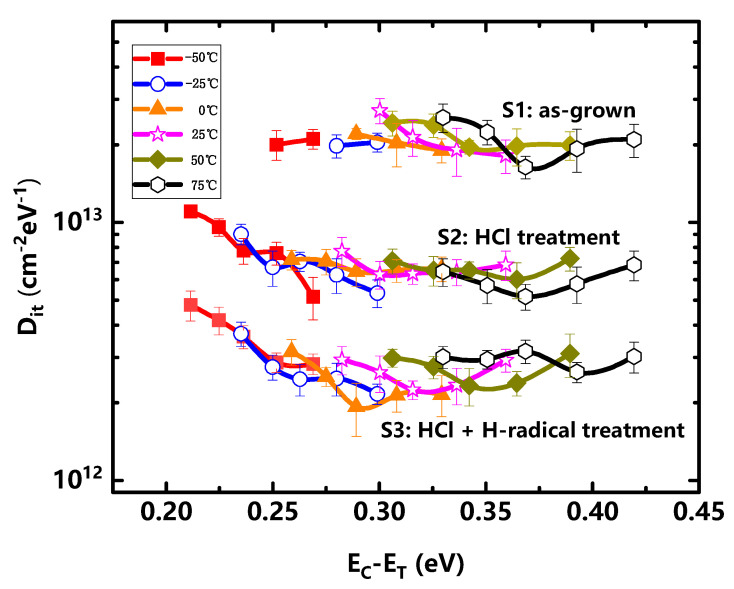
Al_2_O_3_/AlGaN interface $D_{it}$ distribution with different surface treatments.

**Figure 5 micromachines-14-01278-f005:**
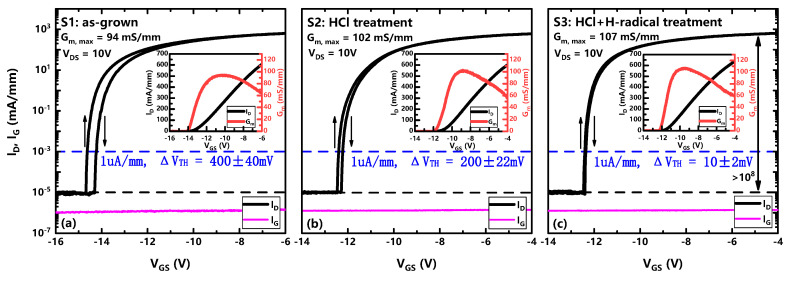
Transfer characteristics of the GaN MIS-HEMT devices (**a**) S1: as-grown; (**b**) S2: HCl treatment; (**c**) S3: HCl + H-radical treatment at 25 °C.

**Figure 6 micromachines-14-01278-f006:**
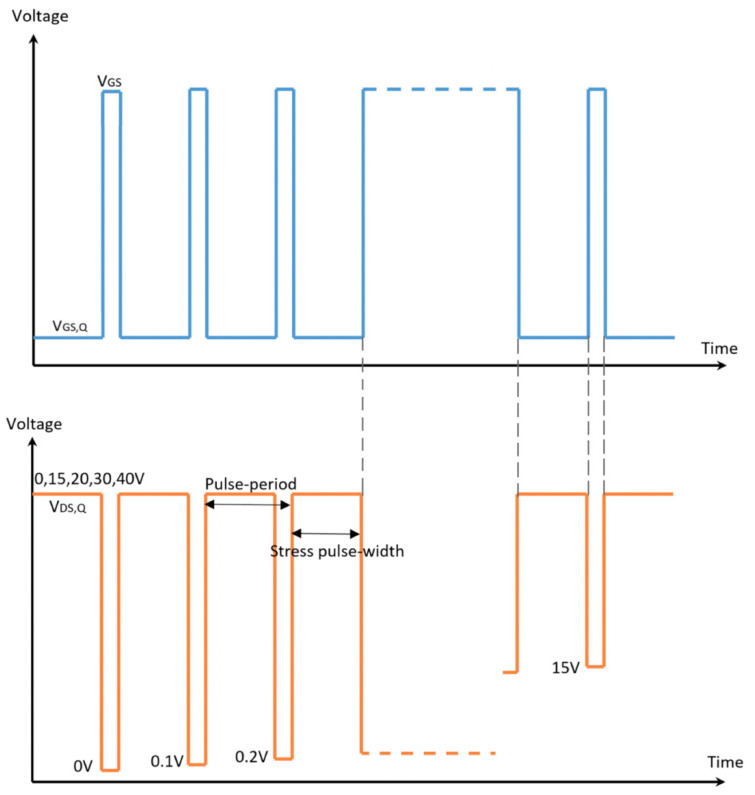
The illustration of the double pulse measurement sequence as a function of time.

**Figure 7 micromachines-14-01278-f007:**
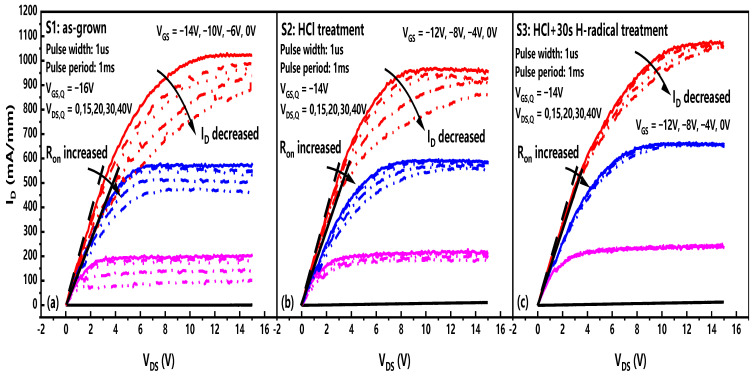
The output characteristics of (**a**) S1: as-grown; (**b**) S2: HCl pretreatment; (**c**) S3: HCl+ H-radical treatment based on the double-pulse tests with different biases at 25 °C.

**Figure 8 micromachines-14-01278-f008:**
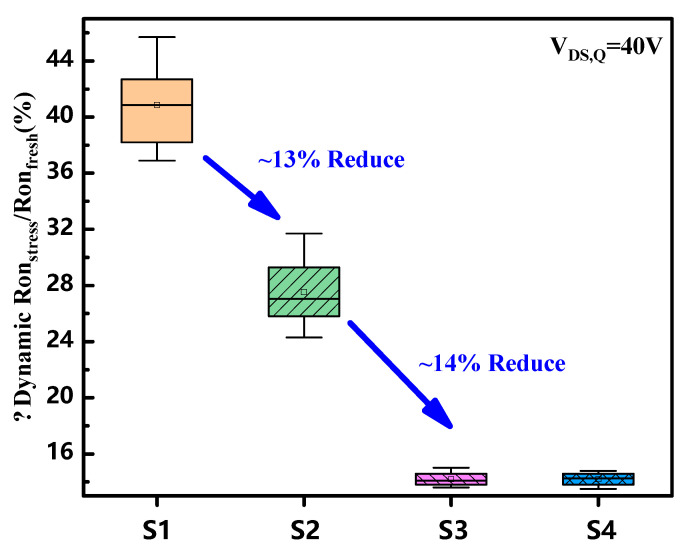
Statistical data of the dynamic $R_{on}$ degradation under the four processes at 25 °C of GaN MIS-HEMTs. The device was stressed under $V_{DS,Q}=40\;V$ and measured at $V_{GS}=0\;V$ with a pulse width of 1 μs.

**Table 1 micromachines-14-01278-t001:** Parameters utilized in the calculation.

Parameters	Value	Unit
$\sigma_{n}$	$1\times 10^{-14}$ [31,33]	${cm}^{2}$
$v_{sat}$	$2\times 10^{7}$ [33]	cm/s
$N_{c}$	$2.7\times 10^{18}$ [33]	${cm}^{-3}$

## Data Availability

Not applicable.

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
