# Peer review of "In Situ H-Radical Surface Treatment on Aluminum Gallium Nitride for High-Performance Aluminum Gallium Nitride/Gallium Nitride MIS-HEMTs Fabrication"

_micromachines, 2023, doi:10.3390/mi14071278_

Round 1

Reviewer 1 Report

see attached pdf file

Author Response

Dear Reviewer, Thank you for taking the time to check our manuscript.We really appreciate your consideration and insightful comments.
We have studied the comments carefully and revised the paper accordingly.
Please see the attachment.
Best regards, Authors of micromachines-2411999 State Key Lab of ASIC and Systems, School of Microelectronics, Fudan University, Shanghai 200433, China

Reviewer 2 Report

After review this work, I can not recommend for publication. There is no novelty in this work.

The state of the art is not well establish, no p-type gate and normally OFF HEMT was discussed. The buffer was not well described just AlGaN but normally a special process is needed to avoid SiN layer. Normally using AlN.

There is no Capacitance Voltage analysis, normally the 2DEG depth profile is discussed.

The transport properties are not well discussed.

The H treatment is not well described.

The conclusion is too pour.

The work must be revised by a native speaker.

Author Response

(The authors gave the same response as above.)

Reviewer 3 Report

Thank you for giving me the opportunity to review this paper. Here some questions and comments that should be addressed by the authors:

·       For the Hydrogen treatment please give more detail about the plasma conditions

·       Please add a band diagram of your device showing the two charges accumulation located in barrier/dielectric and GaN/AlGaN interfaces observed in figure2.

·       The pinch-off voltage of your devices seems to be very high. The expected Vth of your epi-structure should be around -5V even with 25nm of Al2O3 as gate oxide. Please explain why you obtain a Vth in the range of -12 to -14V. 

·       Explain also the impact of the pre-treatment on the Vth.     

Author Response

(The authors gave the same response as above.)

Reviewer 4 Report

In this paper, the in-situ atomic-H pre-treatment was implemented in the fabrication of Al2O3/AlGaN/GaN MIS-HEMTs for the purpose of reducing their current collapse and the on-resistance decay. The content is well organized, however, some crucial issues should be clarified before further consideration.

(1) Throughout the paper, the atomic-H surface treatment was considered to be a highlight to reduce the current collapse of MIS HEMTs, however, the its detailed implementation scheme was not mentioned. The inherent nature of the performance improvement achieved by atomic-H treatment was also not clear.

(2) In this paper, it was lack of necessary description for the epitaxial growth of the AlGaN/GaN heterostructure material.

(3) In Device characterization section, it mentioned that the epitaxial material was composed of 1 nm AlN spacer layer. However, there was ignored in schematic diagram of Figure 1.

Minor editing of English language required

Author Response

(The authors gave the same response as above.)

Round 2

Author Response

Dear reviewer,

We really appreciate your consideration and insightful comments concerning our manuscript entitled “In-situ H-radical surface treatment on AlGaN for high-performance AlGaN/GaN MIS-HEMTs Fabrication” (Manuscript ID:micromachines-2411999). Those comments are very valuable and helpful for improving the quality and readability of our paper. The attachment is our reply letter to you.

Reviewer 2 Report

The work was enhanced. However, I do not see the novelty.

I do not recommend for publication.

Carrier depth profile is needed.

SIMS analysis of the H and other impurities are needed.

XPS near the tramps is needed to see VBM.

Novel publications of HEMTs must be added in the introduction secction.

The english is ok

Author Response

Dear reviewer,

        We really appreciate your consideration and insightful comments concerning our manuscript entitled “In-situ H-radical surface treatment on AlGaN for high-performance AlGaN/GaN MIS-HEMTs Fabrication” (Manuscript ID:micromachines-2411999). Those comments are very valuable and helpful for improving the quality and readability of our paper. The attachment is our reply letter to you.

Authors of micromachines-2411999
State Key Lab of ASIC and Systems, School of Microelectronics, 
Fudan University, Shanghai 200433, China

Reviewer 4 Report

The authors have revised the manuscript according to the comments. I recommand its acception as it is now.

Author Response

Dear reviewer,

        Thank you very much for your positive feedback and the time and effort you have invested in our manuscript.

Authors of micromachines-2411999
State Key Lab of ASIC and Systems, School of Microelectronics, 
Fudan University, Shanghai 200433, China

Round 3

Reviewer 2 Report

Almost all of the comments were attend for the authors.

I recommend the work for publication.

I suggest to review this work by a native speaker.